# Characterization of Insulin-Like Growth Factor Binding Protein 7 (Igfbp7) and Its Potential Involvement in Shell Formation and Metamorphosis of Pacific Abalone, *Haliotis discus hannai*

**DOI:** 10.3390/ijms21186529

**Published:** 2020-09-07

**Authors:** Md. Rajib Sharker, Shaharior Hossen, Ill-Sup Nou, Kang Hee Kho

**Affiliations:** 1Department of Fisheries Science, College of Fisheries and Ocean Sciences, Chonnam National University, 50 Daehak-ro, Yeosu, Jeonnam 59626, Korea; mrsharker@pstu.ac.bd (M.R.S.); 186465@jnu.ac.kr (S.H.); 2Department of Horticulture, College of Life Science and Natural Resources, Sunchon National University, 255, Jungang-ro, Suncheon-Si, Jeollanam-do 57922, Korea; nis@sunchon.ac.kr

**Keywords:** *Haliotis discus hannai*, IGFBP7, mRNA expression, metamorphosis, in situ hybridization

## Abstract

Insulin-like growth factor binding proteins (IGFBPs) are secreted proteins that play an important role in IGF regulation of growth and development of vertebrate and invertebrates. In this study, the IGFBP7 gene was cloned and characterized from mantle tissues of *H. discus hannai*, and designated as Hdh IGFBP7. The full-length cDNA sequence transcribed from the Hdh IGFBP7 gene was 1519-bp long with an open reading frame of 720-bp corresponding to a putative polypeptide of 239 amino acids. The molecular mass of its mature protein was approximately 23.44 KDa with an estimated isoelectric point (pI) of 5.35, and it shared significant homology with IGFBP7 gene of *H. madaka*. Hdh IGFBP7 has a characteristic IGFBP N-terminal domain (22–89 aa), a kazal-type serine proteinase inhibitor domain (77–128), and an immunoglobulin-like C2 domain (144–223). Furthermore, twelve cysteine residues and a signature motif of IGFBPs (XCGCCXXC) were found in its N-terminal domain. Phylogenetic analysis revealed that Hdh IGFBP7 was aligned with IGFBP7 of *H*. *madaka*. Tissue distribution analysis showed that the mRNA of Hdh IGFBP7 was expressed in all examined tissues, with the highest expression level observed in the mantle and gill tissues. The expression level of Hdh IGFBP7 mRNA was relatively higher at the juvenile stage during its metamorphosis period. In situ hybridization showed that Hdh IGFBP7 transcript was expressed in epithelial cells of the dorsal mantle pallial and mucus cells of the branchial epithelium in gill. These results provide basic information for future studies on the role of IGFBP7 in IGF regulation of shell growth, development and metamorphosis of abalone.

## 1. Introduction

Insulin-like growth factors are evolutionarily conserved polypeptides that play a critical role in cell growth, proliferation, differentiation, reproduction, and aging in vertebrates and invertebrates [1,2,3,4,5,6]. The IGF system includes IGF-I, IGF-II, IGF receptors (IGF-IR), and IGF binding proteins (IGFBPs). IGFBPs are involved in controlling and regulating the biological functions of IGFs. In serum and extracellular fluids, IGFBPs can protect IGFs from degradation and modulate their potential biological functions in animals. They also have IGF independent actions via integrin receptors on cell membranes or direct nuclear actions into the nucleus [7]. In vertebrates, six distinct members of IGFBPs (designated as IGFBP 1 to IGFBP 6) have been isolated and characterized [8,9,10]. IGFBPs are a family of secreted proteins that contain a highly conserved N-terminal domain, a conserved C-terminal domain, and a variable central linker (L) domain that plays a crucial role in maintaining their structural integrity [11,12]. Such a domain structure is highly conserved among different vertebrate and invertebrate species [13]. The amino terminus region contains twelve conserved cysteine residues as typical features of the family of IGFBPs. All N-domains possess a signature domain of the IGFBP family (IB domain) required for IGF binding, although their C-domains may also contribute to their ligand-binding capacities [14]. The presence of conserved cysteine residues of their C-domains often mediates the interactions of IGFBPs with other proteins. Conserved cysteine residues in the N-domain and the C-domain are also involved in intradomain disulfide bond formation [15,16]. They help to form the globular structure of N- and C-domains. Their central regions are variable, often containing sites for post-translational regulation, including glycosylation, phosphorylation, and proteolysis. Although different forms of IGFBPs share a common domain organization, each IGFBP type, with its unique biochemical properties, possesses stimulatory or inhibitory functions in different IGF-mediated biological activities [17,18].

IGFBP7 is a secreted protein belonging to the IGFBP family. It can bind strongly to insulin, suggesting that IGFBP7 might have multiple functions than other IGFBPs. It is involved in reproduction, development, differentiation, proliferation, and immunoreaction [9].

Abalone is a highly-priced shellfish ubiquitously distributed throughout temperate and tropical coastal regions [19]. Pacific abalone *H. discus hannai* is an important cultivated abalone species in many Asian countries with a high commercial value because it contains bioactive molecules beneficial for human health [20].

Many experimental studies have been performed on IGFBPs in vertebrate species including *Homo sapiens* [21], *Mus musculus* [22], *Oncorhynchus mykiss* [23], and *Danio rerio* [24]. At present, only a few studies have focused on IGFBPs of mollusks. A full-length cDNA sequence encoding IGFBP7 gene cloned from *H. diversicolor* has been reported to play a crucial role in the immune system of abalone [25]. However, no study has been performed regarding the molecular characterization and expression analysis of the IGFBP7 gene in *H. discus hannai*. Thus, the objective of this study was to obtain the full length cDNA of IGFBP7 from *H. discus hannai* followed by evolutionary analysis. In addition, we determined spatiotemporal expression profiles of IGFBP7 mRNA during metamorphosis. To explore the functional role of Hdh IGFBP7, we also demonstrated the cellular localization of its mRNA by in situ hybridization.

## 2. Results

### 2.1. Cloning and Analysis of IGFBP7

The complete coding sequence of IGFBP7 in *H. discus hannai* was isolated from mantle by 3′-RACE and 5′-RACE PCR and named Hdh IGFBP7. The complete nucleotide sequence of Hdh IGFBP7 (GenBank accession number: MT345605) was 1519-bp in length, including a 137-bp 5′-untranslated region (UTR) and a 662-bp 3′-UTR with a canonical polyadenylation signal sequence (AATAAA) located 10-bp upstream of the poly-A tail. Its open reading frame had a length of 720-bp, encoding 239 deduced amino acids (Figure 1).

The predicted polypeptide sequence contained a putative signal peptide of 18 amino acid residues, resulting in a mature protein of 221 amino acids. The molecular weight and theoretical isoelectric point (pI) of its mature protein were predicted to be 23.44 KDa and 5.35, respectively. Its amino acid composition analysis revealed that glycine was the most abundant amino acid (12.1%), while histidine and tryptophan were the least abundant (1.3%) (Figure 2A,B).

In silico analysis (protcomp; http://www.softberry.com/berry.phtml) indicated that the deduced Hdh IGFBP7 is likely to be an extracellular (secreted) protein. Homology analysis based on its amino acid sequences revealed that the predicted Hdh IGFBP7 displayed significant identities (97.91%) with IGFBP7 of *H. madaka*. However, it shared less amino acid identities with IGFBP7 of piscine and mammalian vertebrates (Table 1).

Sequence analysis of its conserved domains indicates that Hdh IGFBP7 has an IGFBP N-terminal domain (22–89 aa), a kazal-type serine proteinase inhibitor domain (77–128), and an immunoglobulin-like C2 domain (144–223). Characteristic motifs (XCGCCXXC and CGXDXXTYXN) were identified from the deduced amino acid sequence of Hdh IGFBP7. The cloned sequence of Hdh IGFBP7 contained two potential sites for N-linked glycosylation and seven potential sites for phosphorylation by protein kinase C. Moreover, twelve cysteine residues were predicted to form an intramolecular disulfide bond.

Amino acid sequence alignment of IGFBP7 homologs from protostome and deuterostome species is shown in Figure 3.

The output of multiple sequence alignments demonstrates that cysteine residues and the signature domain of the IGFBP family are conserved in abalone. Furthermore, the consensus signature motif that is characteristic of all KI domains of various species is well conserved.

To determine the evolutionary positions of Hdh IGFBP7, a phylogenetic tree was generated with IGFBP proteins of molluscan and selected vertebrate species using the neighbor-joining method (Figure 4).

Phylogenetic analysis showed that the cloned gene of *H. discus hannai* was placed within the IGFBP7 family and phylogenetically most closely related to IGFBP7 of *H. madaka* with a very high bootstrap support value. Serine protease of *H. discus hannai* was used as an outgroup.

Three-dimensional structures of IGFBP7 in *H. discus hannai* and *H. madaka* were constructed using I-Tasser software (Figure 5).

Results indicated that the predicted spatial structure of Hdh IGFBP7 was similar to that of *H. madaka* IGFBP7, both displaying the following three-domain organization: α helix for N- and C-termini domains and β sheets primarily for the mid-domain. Positions of conserved cysteine motifs were also identical to those of known *H. madaka* IGFBP7.

### 2.2. Tissue Expression Analysis of HdhIGFBP7

Relative mRNA expression levels of Hdh IGFBP7 in different tissues of both male and female were analyzed by qRT-PCR (Figure 6).

Results revealed that Hdh IGFBP7 mRNA was expressed in all examined tissues. Hdh IGFBP7 mRNA levels were higher in female than in male. In male, the highest expression level of Hdh IGFBP7 mRNA was found in the mantle. It was significantly higher than that in the cerebral ganglion. Meanwhile, significantly higher expression (*p* < 0.05) level of Hdh IGFBP7 mRNA was detected in the gill compared to that in the cerebral ganglion. In female, Hdh IGFBP7 mRNA showed significantly higher expression in the mantle and gill than in other examined tissues.

Temporal expression of Hdh IGFBP7 mRNA transcript during metamorphosis was analyzed using qRT-PCR assay. Hdh IGFBP7 mRNA was the most abundant at the juvenile stage, although relatively high levels were also detected in late veliger and post-larval stages (Figure 7).

To elucidate the function of Hdh IGFBP7, we also demonstrated the cellular localization of Hdh IGFBP7 mRNA in mantle and gill tissues by in situ hybridization (Figure 8 and Figure 9).

Results showed positive hybridization signals in epidermal cells of dorsal mantle pallial (a region known to express genes involved in the synthesis of the nacreous layer of the shell), and branchial epithelium of gill. No hybridization signal was detected in the mantle or gill tissues when hybridization was performed with a sense probe as a negative control.

## 3. Discussion

IGFBPs are cysteine-rich proteins with conserved N-domains and C-domains. They play important roles in diverse physiological functions in vertebrates and invertebrates. IGFBPs of mammalian vertebrates are well-studied. Several members of the IGFBP superfamily have been documented in invertebrate species, including Py IGFBP in the scallop *Patinopecten yessoensis* [26], Cq IGFBP in the red claw crayfish *Cherax quadricarinatus* [27], Sv IGFBP in the Eastern rock lobster *Sagmariasus verreauxi* [28], and Mn IGFBP in the oriental river prawn *Macrobrachium nipponense* [29]. IGFBP7 has been cloned and characterized in the small abalone (*H. diversicolor*) and named Sa IGFBP7 [25]. In the present study, a full length cDNA sequence of the IGFBP7 gene was isolated for the first time from the Pacific abalone, *H. discus hannai*. The architecture of this cloned sequence displayed several key features of the IGFBPs family, including N-terminal signal peptide, phosphorylation site and IB domain in the N-terminal region. These findings are in agreement with the results of previous reports [30,31]. Potential phosphorylation sites in this cloned sequence might play an indispensable role in cell signaling [32]. There was an 18-aa signal peptide similar to that in Sa IGFBP7, suggesting that Hdh IGFBP7 might be a secreted protein. Twelve conserved cysteine residues allow the formation of intramolecular disulfide bridge, suggesting that they are crucial for three-dimensional conformation of IGFBPs [33] and IGF binding [16,34]. A BLASTP analysis revealed that the deduced amino acid sequence of Hdh IGFBP7 shared substantial sequence identities with IGFBP7 of gastropod molluscan species.

Multiple sequence alignment presented the conserved XGCCXXC domain in its deduced amino acid sequences as typical features of IGFBPs [9,35]. The significance of the XGCCXXC motif is not yet known. It has been hypothesized that this motif of the IB domain plays a role in interaction with IGFs [9]. The single IB domain (SIBD) of IGFBP plays a key role in the endocrine and immune systems of invertebrates [34,36,37]. Li et al. [25] reported that IB domains of Sa IGFBP7 are fundamental for the immune response of invertebrates. Based on the results of previous studies, it is reasonable to assume that Hdh IGFBP7 might be involved in the innate immune system of abalone. The identified Hdh IGFBP7 contains KI and IgC2 domains instead of typical thyroglobulin type-I repeat domain (TY) in the C-terminal region. The C-terminal region of IGFBP (1–6) contains a highly conserved TY domain [9]. In IGFBPs, the role of the TY domain remains unclear, although it is likely to affect IGFs binding and it might participate in the binding of IGFBPs to cell surfaces and/or to extracellular matrix (ECM) proteins via heparin-binding sites [9]. However, to date, very limited information is available about the predominant role of KI and IgC2 domains of IGFBP7. These two domains are distributed in a wide range of multicellular organisms. They play a crucial role in various physiological mechanisms [38,39]. Moreover, the characteristic motif CGXDXXTYXN identified in the KI domain of Hdh IGFBP7 is known to participate in the immune reactions of invertebrates [40]. Further investigations are needed to address the possible involvement of Hdh IGFBP7 in the immune system of abalone.

A phylogenetic tree was constructed with IGFBPs of various species to evaluate their evolutionary relationships. Phylogenetic analysis showed that Hdh IGFBP7 was placed within the IGFBP7 family and phylogenetically clustered with IGFBP7 of *H. madaka* belonging to the family of haliotidae. The sequence similarity, domain conservation, and phylogenetic analysis results indicated that this gene is a potential member of the IGFBP family.

The 3D structure of Hdh IGFBP7 protein was determined based on its amino acid sequence using I-Tasser (Figure 5). The overall structure of Hdh IGFBP7 resembled IGFBP7 of *H. madaka*, showing similarities with α helix and β sheets of *H. madaka*.

qRT-PCR analysis revealed that transcripts of Hdh IGFBP7 were widely expressed in all tested tissues. The ubiquitous expression of Hdh IGFBP7 mRNA transcript in different tissues of abalone is consistent with mammalian and rainbow trout tissue expression patterns of IGFBP7 [23,41]. Transcriptional levels of Hdh IGFBP7 were significantly higher in mantle and gill tissues. Such tissue-specific higher expression in mantle tissue suggests that Hdh IGFBP7 is potentially involved in shell formation, growth, and development. It is well-established that gill is the main entry site of pathogens. Relatively higher expression levels of Hdh IGFBP7 in gills indicate that the involvement of this gene in defense against invading pathogens, in agreement with the results of previous studies on IGFBP7 of *H. diversicolor* [25].

Our ontogenic study of Hdh IGFBP7 expression revealed that its mRNA level was increased gradually before metamorphosis, with the highest level observed in the juvenile stage, suggesting that Hdh IGFBP7 is involved in metamorphosis of *H. discus hannai*. This finding is incongruent with the results of previous reports on *H. diversicolor* [42].

Expression and distribution of IGFBP have been documented in humans [43], rat [44] and zebra fish [45] using an in situ hybridization technique. The localization of Sa IGFBP7 mRNA was analyzed in hemocytes and gill filaments of *H. diversicolor*. To date, cellular localization of IGFBP7 mRNA transcript in mantle or gill tissues of *H. discus hannai* using in situ hybridization has not been reported yet. The mantle is a well-known shell-forming tissue in mollusks and it has been reported that the mantle secretes protein in the epithelial cells to induce shell formation [46,47]. Positive hybridization signal of Hdh IGFBP7 mRNA was detected in the epithelial cells of the dorsal mantle. It is well known that the genes expressed at the mantle pallial are involved in the formation of nacreous layer [48]. Distributions of the Hdh IGFBP7 mRNA gene in the mantle can reflect the involvement of this gene in the shell formation.

In addition, positive signals were found in the branchial epithelium of gills, suggesting that Hdh IGFBP7 might be synthesized in epithelial cells and then mixed with humoral fluid for protecting animals against bacterial infections. The present observation is in accordance with previous reports [25].

## 4. Materials and Methods

### 4.1. Biological Materials and Sample Collection

Two-year-old hatchery propagated male and female abalone were collected and transferred to the laboratory in the Department of Fisheries Science, Chonnam National University, Korea. The hemolymph of abalone was collected from the pericardial sinus using 1 mL syringe and immediately transferred into microtubes followed by centrifugation at 8000× *g* for 5 min at 4 °C to collect the hemocytes. Abalones were dissected on ice to obtain cerebral ganglion, shell muscle, gill, mantle, heart, digestive gland, testis, and ovary tissues. All experimental embryo and larvae were collected in May during the reproductive season. All samples were immediately frozen in liquid nitrogen and stored at −80 °C until total RNA extraction. For in situ hybridization, cryosections were prepared from mantle and gill tissues following a previously reported method by Sharker et al. [49,50,51].

All experiments were conducted after obtaining approval from the Institutional Animal Care and Use Committee of Chonnam National University (CNU IACUC-YS-2020-5).

### 4.2. Isolation of RNA and cDNA Synthesis

Total RNA was extracted from various tissues of Pacific abalone using an RNeasy mini kit (Qiagen, Hilden, Germany) and treated with RNase-free DNase (Promega, Madison, WI, USA) to get rid of genomic DNA contamination. RNA quality and concentration were determined by spectrophotometry (NanoDrop^®^ NP 1000 spectrophotometer, Thermo Fisher Scientific, Waltham, MA, USA). One microgram of total RNA was reverse transcribed using a Superscript^®^ III First-Strand synthesis kit (Invitrogen, Carlsbad, CA, USA) according to the protocol provided by the manufacturer.

### 4.3. Molecular Cloning and Sequencing of IGFBP7

To obtain the full-length cDNA of *H. discus hannai* IGFBP7, a cDNA fragment was amplified with reverse transcription (RT) primers (forward: 5′-AGTGTGACAGAAGTGCCTG-3′ and reverse: 5′-CACAGATCCACCAGTGTAG-3′) designed based on known *H. madaka* IGFBP7 cDNA sequence (GenBank accession no. KP734098.1). One microliter of synthesized cDNA from the mantle was used as a template for RT-PCR in a 20 μL reaction volume composed of 1 μL (20 pmol) each of forward and reverse primers, 4 μL of 5× Phusion HF buffer (1×), 2 μL of dNTP (200 μM), 0.5 μL of 1 U Phusion DNA polymerase and 10.5 μL sterile distilled water (dH_2_O). The amplification reaction included a pre-denaturation step at 95 °C for 3 min, followed by 35 cycles of denaturation at 94 °C for 2 min, annealing at 57 °C for 1 min and extension at 72 °C for 1 min, with a final dissociation step of 5 min at 72 °C.

Amplification products were purified using a gel extraction kit (Promega, Madison, WI, USA), ligated into pTOP Blunt V2 vector (Enzynomics, Daejeon, Korea), and then transformed into competent *Escherichia coli* DH5α cells (Enzynomnics, Daejeon, Korea). A plasmid mini kit (Qiagen, Hilden, Germany) was employed to extract plasmid DNA, which was then sequenced using a Macrogen Online Sequencing System (Macrogen, Seoul, Korea). To obtain the full length sequence, touchdown PCR was conducted with 25 cycles for rapid amplification of 3′ cDNA ends (3′-RACE) and 30 cycles for 5′-RACE PCR following the manufacturer’s instructions. The 50 µL reaction mixture contained 2.5 µL cDNA template from mantle, 5 µL 10 × UPM (Universal Primer A mix), 1 µL gene-specific primer sequences (GSPs), including a 15-bp overlap with the 5′-end of the GSP sequence (antisense primer: 5′-GATTACGCCAAGCTT CCTCCATCTTCAAGCAAATCTCGCAGCA-3′, sense primer: 5′-GATTACGCCAAGCTTGACTGAAGTGCAGCAAGGTGGACGTC-3′), 25 µL SeqAMP buffer, 1 µL SeqAMP DNA polymerase and 15.5 µL PCR grade water. Purified amplification products were ligated into the linearized pRACE vector, transformed into stellar competent cells and sequenced with Macrogen Online Sequencing System (Macrogen, Seoul, Korea). Finally, sequenced RACE products were assembled by overlapping with the initial fragment.

### 4.4. Sequence Characterization and Phylogenetic Analysis of Hdh IGFBP7

The protein domain of the deduced Hdh IGFBP7 gene was analyzed using SMART (http://smart.embl-heidelberg.de/). Its isoelectric point and molecular mass were estimated using ProtParam (http://expasy.org/tools/protparam.html), and subcellular localization was determined with Protcomp (http://linux1.softberry.com/berry.phtml). Its potential N-linked glycosylation and serine/threonine phosphorylation sites were predicted using NetNGlyc 1.0 Server and NetPhosK 3.1 Server, respectively. Disulfide bonding state of cysteines and N-terminal signal peptide in this protein sequence was determined using CYSPRED [52] and SignalP 4.1 [53], respectively. Multiple sequence alignment of Hdh IGFBP7 was performed using Clustal Omega [54].

To construct a phylogenetic tree, IGFBP protein homologues from protostomes and deuterostomes were retrieved from the NCBI database and aligned using Clustal Omega [54]. Phylogenetic and molecular evolutionary analyses were conducted using MEGA 7 with the neighbor-joining algorithm [55]. Bootstrap values were replicated 1000 times to obtain confidence values of nodes.

### 4.5. Homology Modeling of Hdh IGFBP7

Three-dimensional (3D) structure of IGFBP7 from *H. discus hannai* and *H. madaka* were constructed using the I-TASSER server [56]. Predicted 3D structure of Hdh IGFBP7 was visualized and analyzed with Chimera (https://www.cgl.ucsf.edu/chimera/). 

### 4.6. Quantitative Real-Time PCR (qRT-PCR) Analysis

qRT-PCR was performed on a LightCycler^®^ 96 system (Roche, Germany) using a 2× qPCRBIO SyGreen Mix Lo-Rox kit (PCR Biosystems, Ltd., London, UK) with gene-specific primers (forward: 5′-CCACAAACTGAAGCCTGAGG-3′ and reverse: 5′-AGGGTCCAGACTTCTCAATC-3′). Ribosomal protein L-5 (JX002679.1) primer (forward: 5′-TGTCCGTTTCACCAACAAGG-3′ and reverse: 5′-AGATGGAATCAAGTTTCAATT-3′) from *H. discus hannai* was adopted as an internal control based on its expression stability [57]. In qRT-PCR, 1 μL cDNA from cerebral ganglion, shell muscle, gill, mantle, heart, digestive gland, hemocyte, testis, and ovary was used as a template for 20 µL reaction as described by Sharker et al. [50]. PCR was carried out using the following cycling parameters: pre-incubation at 95 °C for 3 min, followed by three-step amplification at 94 °C for 2 min, 60 °C for 1 min, and 72 °C for 30 s for 40 cycles. The experiment was performed with three biological replicates to avoid measurement and operation errors. The melting temperature was used as a default setting. Relative gene expression levels were calculated using the 2^−ΔΔct^ method [58].

### 4.7. In Situ Hybridization (ISH)

The cDNA fragments corresponding to nucleotides (138–1073) of Hdh IGFBP7 transcript were amplified, ligated into pGEM-T Easy vector (Enzynomics, Daejeon, Korea), and sequenced to confirm the identity and orientation of the product. The recombinant plasmid was used as a template to synthesize antisense and sense RNA probes by in vitro transcription using a DIG RNA Labeling kit (Roche Diagnostics, Germany) and T7 or SP6 RNA polymerase following the manufacturer’s instruction. In situ hybridization was performed according to the method described by Sharker et al. [59,60]. Mantle and gill tissue sections were hybridized with DIG-labeled antisense or sense RNA probe at 65 °C. Subsequently, these sections were thoroughly washed and treated with blocking solution (10% calf serum in PBST) at room temperature for 1 h followed by incubation with alkaline phosphatase-conjugated anti-digoxigenin-Ap-Fab fragments antibody (diluted 1:500 in blocking solution [Roche]) overnight at 4 °C to detect the hybridization signal. Finally, these sections were treated with BCIP/NBT (Roche) substrate in a dark and humid chamber for at least 1 h. After satisfactory color development, sections were observed using a stereo microscope (SMZ1500, Nikon, Tokyo, Japan) equipped with a digital camera.

### 4.8. Statistical Analysis

Relative mRNA expression levels were subjected to a one-way analysis of variance (ANOVA), followed by Tukey’s multiple comparisons to determine whether there were any significant differences using SPSS (Version 17.0., SPSS Inc., Chicago, IL, USA). Data from qRT-PCR are expressed as mean ± SE and differences were regarded as significant at *p* < 0.05.

## 5. Conclusions

In conclusion, a 1519-bp sequence encoding the IGFBP7 gene was cloned from the mantle tissue and its spatiotemporal expression profile was analyzed in the Pacific abalone *H. discus hannai*. Results of phylogenetic analysis and structural features will contribute to gain further insight into the evolutionary processes of IGFBPs in invertebrates. The highest level of Hdh IGFBP7 mRNA was found in the juvenile stage suggest that this gene is involved in the metamorphosis of abalone by mediating actions of IGFs. In situ hybridization results of mantle and gill tissue suggest that it might play an important role in shell formation, growth, development, and immune responses.

## Figures and Tables

**Figure 1 ijms-21-06529-f001:**
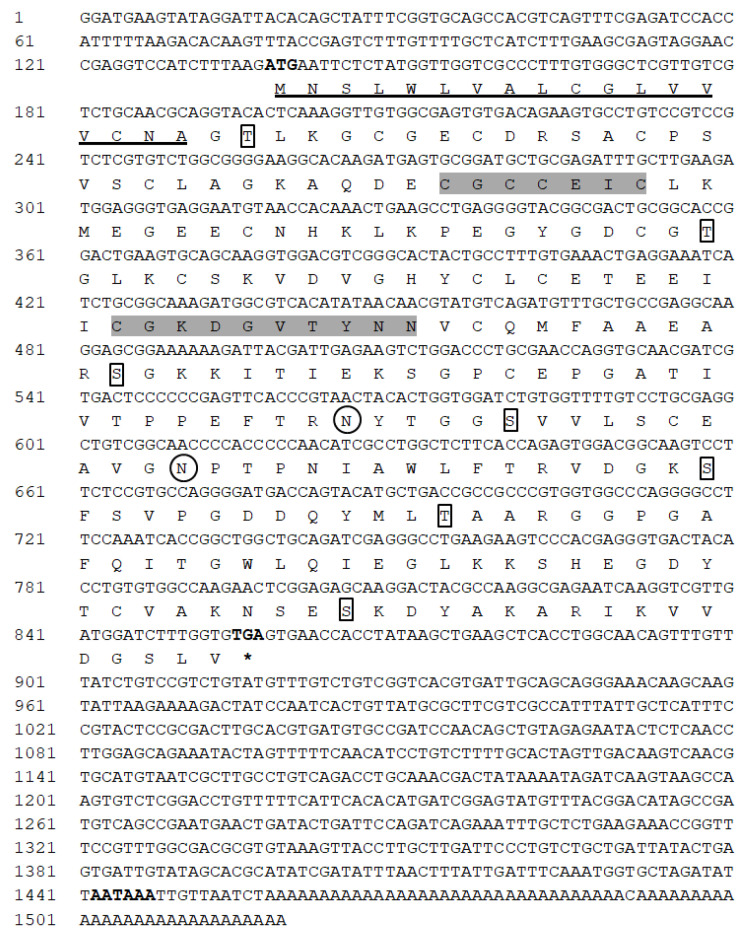
The full-length nucleotide and deduced amino acid sequences of Hdh IGFBP7. The initiation codon, termination codon (asterisks) and putative polyadenylation signal sequence are marked in boldface. The N-linked glycosylation site is circled. Phosphorylation sites are boxed. The signal peptide is underlined. Characteristic conserved motifs “XCGCCXXC” and “CGXDXXTYXN” are highlighted in light grey.

**Figure 2 ijms-21-06529-f002:**
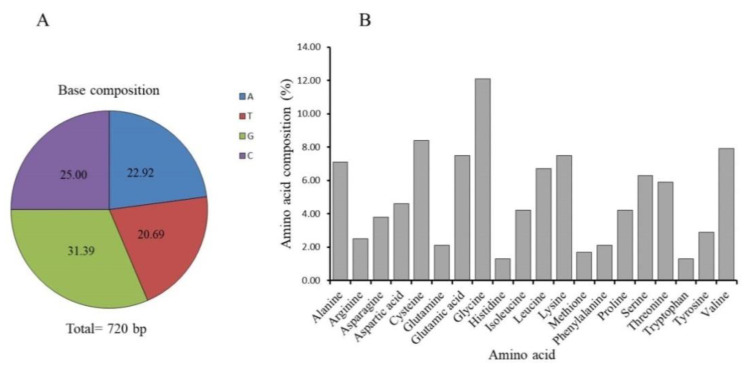
Sequence compositions of Hdh IGFBP7 CDs region at the nucleotide and amino acid levels. (**A**) Base composition, (**B**) Amino acid composition.

**Figure 3 ijms-21-06529-f003:**
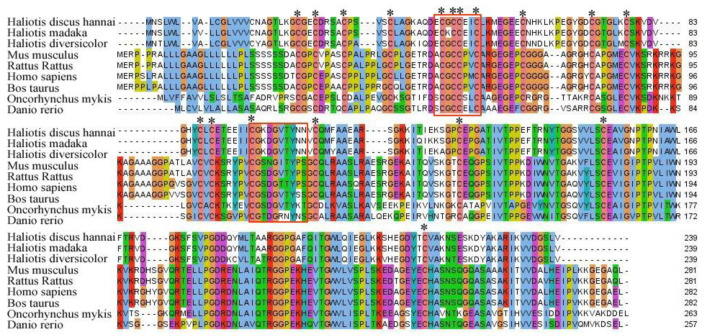
Multiple alignment of deduced amino acid sequences of Hdh IGFBP7 and IGFBP7 of other invertebrate and vertebrate species. Conserved cysteine residues are marked by asterisks. Highly conserved motifs are indicated by red boxes. Hdh, *H. discus hannai*; Hm, *H. madaka*; Hd, *H. diversicolor*; Hs, *Homo sapiens*; Mm, *Mus musculus*; Bt, *Bos taurus*; Dr, *Danio rerio*; Om, *Oncorhynchus mykiss*; Rr, *Rattus rattus*.

**Figure 4 ijms-21-06529-f004:**
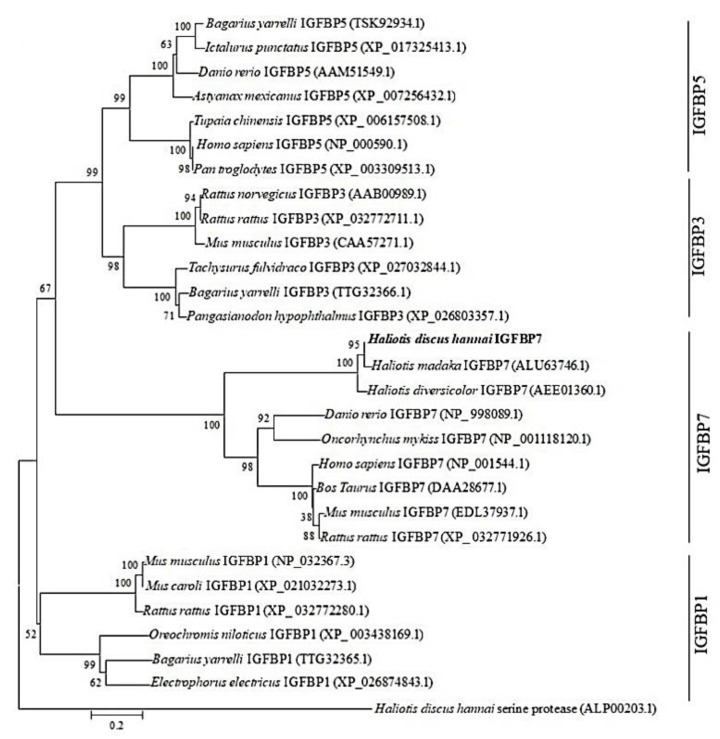
Phylogenetic tree based on amino acid sequences of IGFBPs from vertebrates and invertebrates. Evolutionary analysis was performed using a neighbor-joining method with 1000 bootstrap replicates. The scale bar represents the number of amino acid substitutions per site. All sequences were retrieved from the GenBank of NCBI. GenBank accession numbers are shown in parentheses. Hdh IGFBP7 is highlighted in bold font.

**Figure 5 ijms-21-06529-f005:**
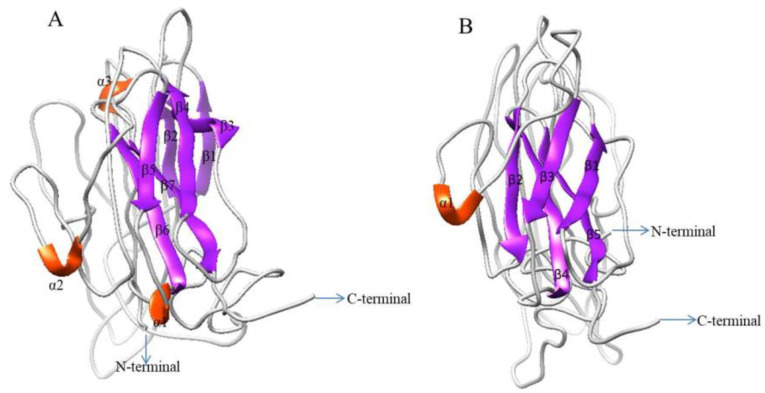
Comparison of 3D structure of IGFBP7 from (**A**) *H. discus hannai* and (**B**) *H. madaka*. N- and C- termini are marked with arrows. The structure was generated using I-TASSER server with a C-score between 2 and 4.

**Figure 6 ijms-21-06529-f006:**
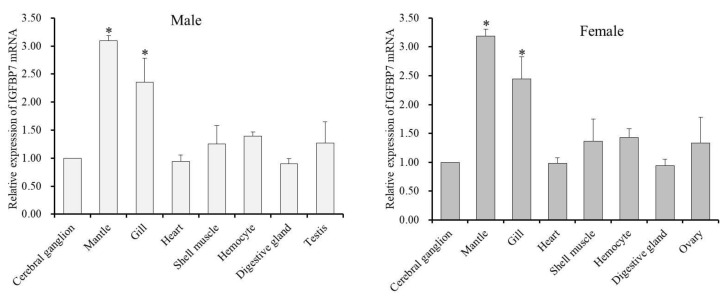
Distribution of Hdh IGFBP7 mRNA in various tissues of *H. discus hannai*. Expression levels of Hdh IGFBP7 mRNA in all tissues were compared with the relative value of the cerebral ganglion (1). Vertical bars represent means ± SE. Asterisks indicate significant differences (*p* < 0.05).

**Figure 7 ijms-21-06529-f007:**
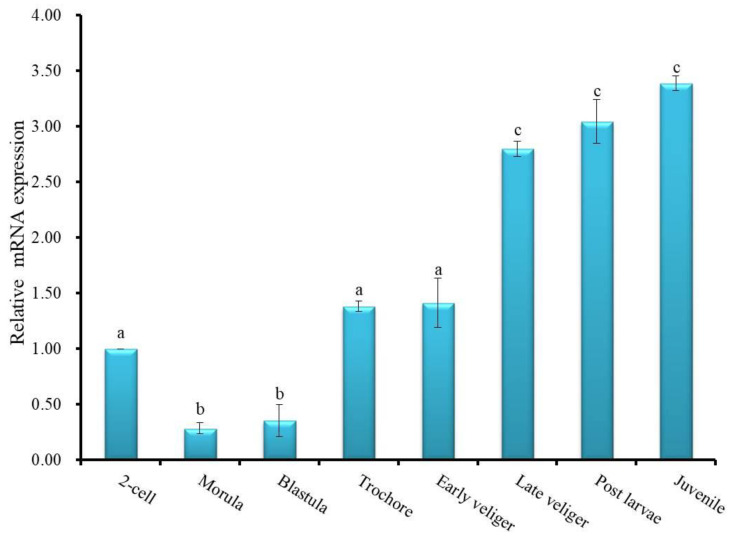
Relative expression levels of Hdh IGFBP7 mRNA in different developmental stages of *H. discus hannai*. Expression values of Hdh IGFBP7 mRNA were normalized to expression in the 2-cell stage (1). Means that do not share the same letter are significantly different.

**Figure 8 ijms-21-06529-f008:**
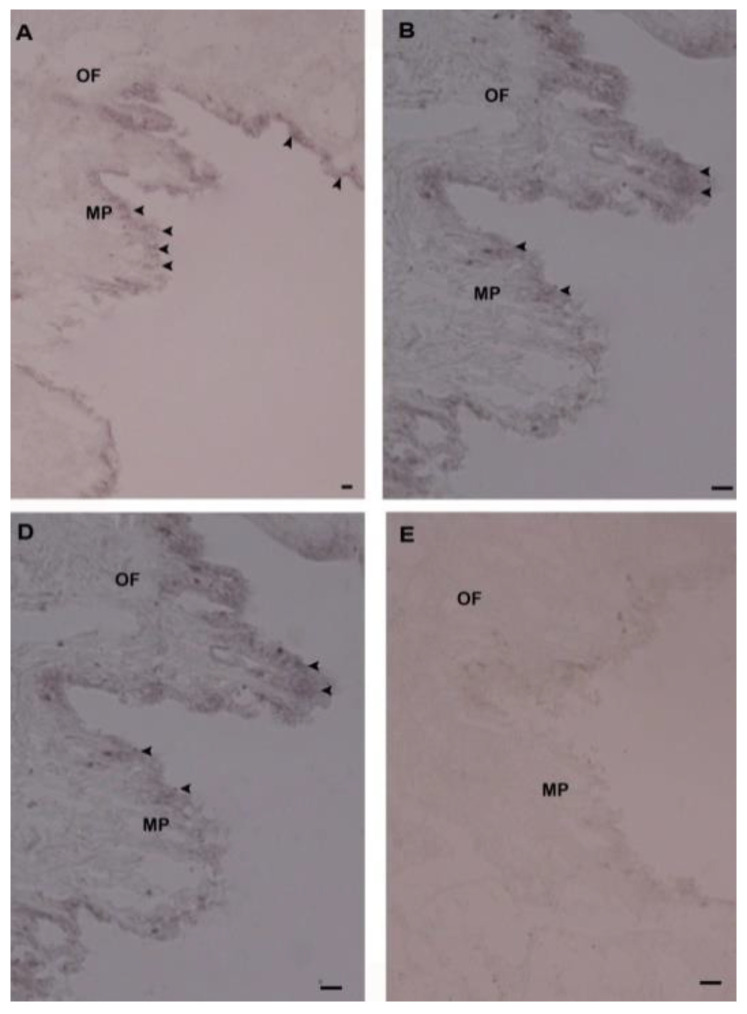
Detection of Hdh IGFBP7 mRNA in the mantle of *H. discus hannai* by in situ hybridization. (**A**) Strong hybridization signals with an antisense probe were found in the outer epithelium of mantle pallial (MP) edge, (**B**) Medium magnification of A, (**D**) High magnification showing hybridized Hdh IGFBP7 mRNA in mantle pallial (MP) edge, (**E**) No hybridization signals were observed in the control section stained with a sense probe. Strong hybridization signals are indicated with black arrowheads. Scale bar, 100 µm. OF, outer fold; MP, mantle pallial.

**Figure 9 ijms-21-06529-f009:**
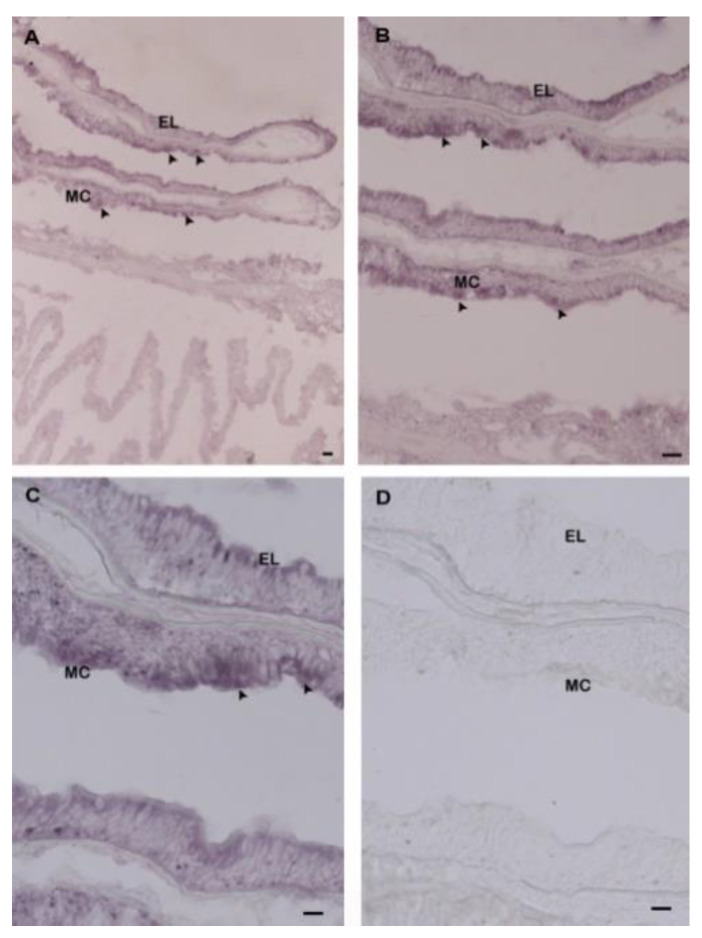
Hdh IGFBP7 mRNA detection in the gill of *H. discus hannai* by in situ hybridization. (**A**) Strong expression of Hdh IGFBP7 with an antisense probe was evident in mucus cells of the branchial epithelium, (**B**) Medium magnification of A; (**C**) High magnification showing hybridized Hdh IGFBP7 mRNA in mucus cells. (**D**) No hybridization signal was seen in the control section stained with a sense probe. Hybridization signals are indicated with black arrowheads. Scale bar, 100 µm. EL, epithelial layer; MC, mucus cells.

**Table 1 ijms-21-06529-t001:** Pairwise comparison (% of identity) of Hdh IGFBP7 with IGFBP7 of other gastropod mollusks, eutherian mammals, and piscine vertebrates.

1	2	3	4	5	6	7	8	9	
	91.21	89.56	94.51	49.75	52.98	39.63	37.80	36.93	1. Human
		97.15	92.89	55.22	51.57	36.40	36.13	35.60	2. Rat
			91.30	54.10	51.18	35.60	35.29	34.80	3. Mouse
				56.34	53.67	36.90	37.08	36.51	4. Cow
					65.86	36.14	35.68	36.12	5. Rainbow trout
						40.41	39.21	39.65	6. Zebra fish
							91.21	93.31	7. Small abalone
								97.91	8. Giant abalone
									9. Pacific abalone

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
