# Peer review of "Characterization of Insulin-Like Growth Factor Binding Protein 7 (Igfbp7) and Its Potential Involvement in Shell Formation and Metamorphosis of Pacific Abalone, *Haliotis discus hannai"

_ijms, 2020, doi:10.3390/ijms21186529_

Round 1

Reviewer 1 Report

This manuscript describes characterization of IGFBP7 and its potential involvement in shell formation and metamorphosis of Pacific abalone. This study is important to understand the role of IGFBP in IGF regulation in this species. Thus, this study may stimulate interest into investigations of shell formation and metamorphosis by growth stimulating peptide in this species.  

Author Response

Comments and Suggestions for Authors

This manuscript describes characterization of IGFBP7 and its potential involvement in shell formation and metamorphosis of Pacific abalone. This study is important to understand the role of IGFBP in IGF regulation in this species. Thus, this study may stimulate interest into investigations of shell formation and metamorphosis by growth stimulating peptide in this species.  

Submission Date

11 August 2020

Date of this review

28 Aug 2020 12:26:03

Reply: Thank you very much for your kind review. Based on your valuable suggestions we have improved the conclusion section.

Reviewer 2 Report

The manuscript by Md Rajib Sharker et al. entitled “Characterization of insulin-like growth factor binding protein 7 (IGFBP7) and its potential involvement in shell formation and metamorphosis of Pacific abalone, Haliotis discus hannai”, presents the cloning and the further characterization of IGFBP7 in several tissues and life-stages of the abalone Haliotis discus hannai, and provides evidence of the ubiquitous presence of this proteins across several tissues, mainly gill and mantle, with increasing expression through ontogeny. Overall, this is a comprehensive characterization of the IGFBP7 based on solid methods. The study is, for the most part, well designed and straightforward, and complements a previous paper by the same authors describing another growth factor in the Pacific abalone (IGFBP5). Thus, the study will be of interest to researchers working with growth and development in marine invertebrates. However, there are some issues that need to be addressed before publication in the journal.

The title of the manuscripts clearly suggests that IGFBP7 is involved in shell formation and metamorphosis, however, through the manuscript and especially in results and discussion sections, the authors describe a rather broad spectrum of functions of this protein, presenting weak evidence of its function for shell formation. I would suggest the authors include more evidence to sustain this conclusion or adjust the title accordingly.

In this regard, a previous study by the same authors characterizes the IGFBP5 in the Pacific abalone. While this study is briefly mentioned in this manuscript, I would encourage the authors to take it on board. Describing the contrasting pattern observed through ontogeny and between tissues would provide more evidence for the different functions of IGBPF7 during development and will be helpful for future studies with this family of proteins.

Specific comments:

Line 95: Provide more details on this statement.

Table I: Indicate the units of the values (% of identity)

Line 150 – 151: The higher value of females than males should be supported by a statistical test. Perhaps a two-way ANOVA would suit this purpose. 

Line 331: Specify how many samples per tissue/life-stage were used for the analysis.

Author Response

Comments and Suggestions for Authors

The manuscript by Md Rajib Sharker et al. entitled “Characterization of insulin-like growth factor binding protein 7 (IGFBP7) and its potential involvement in shell formation and metamorphosis of Pacific abalone, Haliotis discus hannai”, presents the cloning and the further characterization of IGFBP7 in several tissues and life-stages of the abalone Haliotis discus hannai, and provides evidence of the ubiquitous presence of this proteins across several tissues, mainly gill and mantle, with increasing expression through ontogeny. Overall, this is a comprehensive characterization of the IGFBP7 based on solid methods. The study is, for the most part, well designed and straightforward, and complements a previous paper by the same authors describing another growth factor in the Pacific abalone (IGFBP5). Thus, the study will be of interest to researchers working with growth and development in marine invertebrates. However, there are some issues that need to be addressed before publication in the journal.

The title of the manuscripts clearly suggests that IGFBP7 is involved in shell formation and metamorphosis, however, through the manuscript and especially in results and discussion sections, the authors describe a rather broad spectrum of functions of this protein, presenting weak evidence of its function for shell formation. I would suggest the authors include more evidence to sustain this conclusion or adjust the title accordingly.

In this regard, a previous study by the same authors characterizes the IGFBP5 in the Pacific abalone. While this study is briefly mentioned in this manuscript, I would encourage the authors to take it on board. Describing the contrasting pattern observed through ontogeny and between tissues would provide more evidence for the different functions of IGBPF7 during development and will be helpful for future studies with this family of proteins.

Reply: We are thankful for your valuable suggestions. The mantle is well- known shell forming tissues in molluscs. It has been reported that mantle secretes protein in the epithelial cells to induce shell formation (Jablonski, et al. 1990, Werner et al., 2013).

Positive hybridization signal of Hdh IGFBP7 mRNA was detected in the epithelial cells of the dorsal mantle pallial, a region known to express genes involved in the synthesis of the nacreous layer of the shell. It is well known that the genes expressed at the mantle pallial are involved in the formation of nacreous layer (Takeuchi et al. 2013). Distributions of Hdh IGFBP7 mRNA gene in the mantle can reflect the involvement of this gene in the shell formation.

Based on your valuable suggestion, we modified the conclusion section.

References

Takeuchi, T.; Endo, K. Biphasic and dually coordinated expression of the genes encoding major shell matrix proteins in the pearl oyster Pinctada fucata. Mar. Biotechnol. 2006, 8, 52–61.

Jablonski, D.  On Biomineralization . Heinz A. Lowenstam , Stephen Weiner . J. Geol. 1990, 98, 977–977.

Werner, G.D.A.; Gemmell, P.; Grosser, S.; Hamer, R.; Shimeld, S.M. Analysis of a deep transcriptome from the mantle tissue of Patella vulgata Linnaeus (Mollusca: Gastropoda: Patellidae) reveals candidate biomineralising genes. Mar. Biotechnol. 2013, 15, 230–243.

Specific comments:

 Line 95: Provide more details on this statement.

Reply: We are thankful for your valuable suggestions. Based on your valuable suggestion we omitted the previous sentence and incorporated the following sentence in the line number 95-97.

 ‘In silico analysis (protcomp, http://www.softberry.com/berry.phtml) indicated that the deduced Hdh IGFBP7 is likely to be an extracellular (secreted) protein’.

Table I: Indicate the units of the values (% of identity)

Reply: Thank you very much for your kind suggestion. According to your valuable suggestions we incorporated it in Table 1.

Line 150 – 151: The higher value of females than males should be supported by a statistical test. Perhaps a two-way ANOVA would suit this purpose. 

Reply: Thank you very much for your valuable comments and suggestion. We performed qPCR analysis using different tissues of male and female abalone separately and analyzed the result using the 2−ΔΔct method. Then statistical analyzed was done using one-way analysis of variance (ANOVA), followed by Tukey’s multiple comparisons using SPSS (version 16.0) to assess the differences in relative mRNA expression levels.

Line 331: Specify how many samples per tissue/life-stage were used for the analysis.

Reply: We are thankful for your valuable comments. In this study, we used 10 individuals (5 male and 5 female) for various tissue collection. During the collection of different stages of sample, we collect the samples in 1.5 ml microtube after fertilization. 1.5 ml microtube contains about 600 samples. Then we examined the stages using microscope and centrifuged to remove the water. After that the samples transferred in liquid nitrogen before storage at −80 ◦C for subsequent experiment.

Thanks again for your review and kind suggestion.